# Computational Ghost Imaging Based on Light Source Formed by Coprime Array

**DOI:** 10.3390/s20164508

**Published:** 2020-08-12

**Authors:** Yapeng Zhan, Jiying Liu, Zelong Wang, Qi Yu

**Affiliations:** College of Liberal Arts and Sciences, National University of Defense Technology, Changsha 410073, China; yapeng.zhan@nudt.edu.cn (Y.Z.); zelong_wang@163.com (Z.W.); yuqi10@nudt.edu.cn (Q.Y.)

**Keywords:** computational ghost imaging, coprime array, Eisenstein integer

## Abstract

In computational ghost imaging, a spatial light modulator (SLM) can be used to modulate the light field. The relative locations and the number of light point pixels on an SLM affect the imaging quality. Usually, SLMs are two-dimensional arrays which are drawn uniformly or are randomly sparse. However, the patterns formed by a uniform array are periodic when the number of light point pixels is small, and the images formed by a random sparse array suffer from large background noise. In this paper, we introduce a coprime array based on the Eisenstein integer to optimize the light point pixel arrangement. A coprime array is widely used as a microwave radar receiving array, but less implemented in optics. This is the first time that a coprime array based on Eisenstein integer has been introduced in computational ghost imaging. A coprime array with this structure enhances the imaging quality when limited measurements are recorded, and it reduces background noise and avoids periodicity. All results are verified by numerical simulation.

## 1. Introduction

Ghost imaging mainly uses a second-order correlation (intensity fluctuation correlation) of the light field, and enjoys good performance when encountering atmospheric turbulence and scattering effects [1,2,3,4]. It has been widely used in remote sensing [5], optical encryption [6,7], single-pixel imaging [8,9,10], and other fields [11,12].

In the early stages, a quantum entangled light source was used for ghost imaging [13]. In 2004, Gatti et al. [14,15] theoretically proved that ghost imaging could be realized by pseudo-thermal light. Since then, ghost imaging based on pseudo-thermal light has been widely studied in [16,17,18]. In their experiments, the pseudo-thermal light was often obtained by irradiating a rotating ground glass with a laser. Whether in quantum ghost imaging or classical ghost imaging, the imaging system consisted of two optical paths, where the object and a bucket detector without spatial resolution were placed in the test optical path, and a high-resolution charge-coupled device (CCD) was placed in the reference optical path [19]. In 2008, Shapiro [20] first proposed the theory of computational ghost imaging. He proposed the use of a spatial light modulator (SLM) instead of a rotating ground glass and applied deterministic modulation to the laser beam with an SLM. In this way, the intensity fluctuation pattern could be calculated by the diffraction theory, and the CCD was no longer needed. In 2009, Bromberg [21] realized the first computational ghost imaging system in experiments. The computational ghost imaging system no longer needed the reference optical path, which greatly reduced the complexity and was convenient for practical application as compared with the original ghost imaging system.

The relative locations and the number of light point pixels on an SLM affect imaging quality. The simplest way is to arrange the light point pixels of the SLM into a uniform array. However, in this structure, in order to obtain better image quality, a large number of light point pixels is needed. When the number of light point pixels is small, periodic patterns appear and result in aliasing of the target image [22]. Then, random sparse arrays were introduced into the arrangement of light point pixels on an SLM [23]. When light point pixels were arranged randomly, the periodicity of patterns could be avoided, but the images suffer from large background noise. In this paper, the arrangement of light point pixels on the SLM is optimized based on a coprime array. Coprime arrays are widely used in array signal processing [24,25,26], but less implemented in optics. In this paper, we introduce a coprime array into computational ghost imaging for the first time. The coprime array is a type of sparse array, when the number of elements is the same, it has a larger array aperture that reduces the mutual coupling effect between the elements, and therefore improves the resolution and reduces the background noise. In addition, the coprime array breaks through the limit of the number of physical elements on the degree of freedom, which also helps to improve the spatial resolution [27,28]. Therefore, the arrangement of light point pixels on the SLM, according to this structure, can achieve better imaging results than uniform arrays and random sparse arrays.

The rest of the paper is organized as follows: In Section 2, we provide the formulation of coprime array; in Section 3, we present the image reconstruction method based on coprime array; then, in Section 4, we show our numerical simulations; and finally, in Section 5, we draw our conclusions.

## 2. Basic Principle of a Coprime Array

The traditional coprime planar array is a combination of two uniform plane subarrays [29,30]. The first subarray contains M×M elements and the second subarray contains N×N elements, where *M* and *N* are coprime integers. The distance between the adjacent elements of the first subarray is (N×λ)/2, and the distance between the adjacent elements of the second subarray is (M×λ)/2, where *λ* denotes the wavelength of the detection light field. When M=3 and N=4, the structure of the coprime planar array is as shown in Figure 1 [29,30]. In this paper, we choose a new coprime array which is constructed in the Eisenstein domain. The coprime array constructed in Eisenstein domain also consists of two subarrays, but its elements are arranged into hexagons. This is the most densely packed array on a two-dimensional plane, and the degree of freedom can be increased by 15.5% [31] when the actual size of the array is fixed.

ℤ and ℚ denote rational integers {⋯,−1,0,1,⋯} and rational numbers {a/b|a,b∈ℤ, b≠0}, respectively. An ideal Ι is a subset of ℤ[q]={a+bq,a,b∈ℤ} such that whenever x∈Ι and m∈ℤ[q], mx belongs to Ι. If the ideal is generated by a single element in ℤ[q], this ideal is called a principal ideal. In a principal ideal domain where every ideal is principal, prime ideals are simply generated by prime elements. A principal ideal domain ℚ(D) is an Eisenstein domain ℤ[ω] when D=−3, *D* is a square-free rational integer, and ω=eiπ/3=1/2+(3/2)i. The Eisenstein integer can be expressed in the form of a+bω, where a,b∈ℤ. The position of the elements in the subarray is given by the following steps [31]:
Calculate the prime p∈ℤ that satisfies: x2≡−3(modp),x∈ℤ, and here we take p=7,x=2.For the principal ideal domain ℚ(D), if D≡1(mod4), then B=−1 and C=(1−D)/4; conversely, B=0 and C=−D. Therefore, for the Eisenstein domain ℚ(−3), B=−1,C=(1−D)/4=1.Decompose principal idea 〈p〉 into two coprime ideals p1=〈m1+m2ω〉 and p2=〈n1+n2ω〉, where m1,m2,n1,n2 satisfies m12+m1m2+m22=n12+n1n2+n22=p. Here we get p1=〈2+3i〉=〈1+2ω〉 and p2=〈2−3i〉=〈3−2ω〉.The generator matrix corresponding to ideal p1 is:G1=G(m1−Cm2m2m1−Bm2),
and the generator matrix corresponding to ideal p2 is: G2=G(n1−Cn2n2n1−Bn2).If D<0,
G=(1Re(q)0Im(q)),
so, the corresponding generator matrix of idea $p_{1}=\langle 1+2\omega \rangle$ in this paper is:$$G_{1}=\left(\begin{array}{cc} 1 & \frac{1}{2}\\ 0 & \frac{\sqrt{3}}{2} \end{array}\right)\left(\begin{array}{cc} 1 & -2\\ 2 & 3 \end{array}\right)=\left(\begin{array}{cc} 2 & -\frac{1}{2}\\ \sqrt{3} & \frac{3\sqrt{3}}{2} \end{array}\right),$$
the corresponding generator matrix of idea $p_{2}=\langle 3-2\omega \rangle$ is:$$G_{2}=\left(\begin{array}{cc} 1 & \frac{1}{2}\\ 0 & \frac{\sqrt{3}}{2} \end{array}\right)\left(\begin{array}{cc} 3 & 2\\ -2 & 1 \end{array}\right)=\left(\begin{array}{cc} 2 & \frac{5}{2}\\ -\sqrt{3} & \frac{\sqrt{3}}{2} \end{array}\right).$$The position of the elements in the first subarray is: $$G_{1}\left(\begin{array}{l} x_{i}\\ y_{ij} \end{array}\right),$$
where i=1,2,3,4,5;j=1,2,3,4,5. The position of the elements in the second subarray is: $$G_{2}\left(\begin{array}{l} x_{i}\\ y_{ij} \end{array}\right),$$
where i=1,2,3,4,5;j=1,2,3,4,5. The values of *x_i_* and *y_ij_* take the integers in the interval [−2,2], and the two subarrays each have 25 elements. The distance between adjacent elements in the first subarray is: $$\left\|G_{1}\left(\begin{array}{l} 1\\ 0 \end{array}\right)\right\|=\left\|G_{1}\left(\begin{array}{l} 0\\ 1 \end{array}\right)\right\|=\sqrt{7},$$
and the distance between adjacent elements in the second subarray is:$$\left\|G_{2}\left(\begin{array}{l} 1\\ 0 \end{array}\right)\right\|=\left\|G_{2}\left(\begin{array}{l} 0\\ 1 \end{array}\right)\right\|=\sqrt{7}.$$
When the values of *x_i_* and *y_ij_* are both 0, we obtain: $$G_{1}\left(\begin{array}{l} x_{i}\\ y_{ij} \end{array}\right)=\left(\begin{array}{l} 0\\ 0 \end{array}\right),$$
$$G_{2}\left(\begin{array}{l} x_{i}\\ y_{ij} \end{array}\right)=\left(\begin{array}{l} 0\\ 0 \end{array}\right),$$
the positions of the two elements coincide.

We represent the subarray obtained from the generating matrix **G_1_** as **H_1_** and the subarray obtained from the generating matrix **G_2_** as **H_2_**. The distribution of array elements is shown in Figure 2. When considering **H_1_** and **H_2_** separately, the two subarrays can be regarded as two uniform arrays, with the distance between adjacent elements being $\sqrt{7}$ and the furthest distance from the point in the array to the coordinate axis being 9 (see Figure 2). If the light point pixels with phase modulation are distributed on an L×L area and the distance between the two adjacent light point pixels is *d*, then *d* should satisfy the formula $\sqrt{7}/9=d/\left(L/2\right)$, that is, $d=\sqrt{7}L/18$.

## 3. Computational Ghost Imaging Intensity Correlation Theory

The light field distribution at the SLM is:(1)$$E(\vec{r},t)=\sum_{ij}\exp[-{\left(\vec{r}-{\vec{r}}_{ij}\right)}^{2}/r_{c}^{2}]\exp[i\phi_{ij}(t)],$$
where the ${\vec{r}}_{ij}$ is the position of light point pixel *ij* on SLM, *r_c_* is the radius of the light beam, and $\phi_{ij}(t)$ is the preset phase at time *t*.

After a certain distance *z*, the light field distribution becomes:(2)$$E({\vec{r}}^{\prime },t)=\int E(\vec{r},t-z/c)h(\vec{r},{\vec{r}}^{\prime })d\vec{r},$$
where $h(\vec{r},{\vec{r}}^{\prime })$ is the light field transfer function. According to the Fresnel approximation, the light field transfer function in free space is:(3)$$h(\vec{r},{\vec{r}}^{\prime })=\frac{\exp(ikz)}{i\lambda z}\exp[\frac{i\pi }{\lambda z}{\left(\vec{r}-{\vec{r}}^{\prime }\right)}^{2}].$$

According to Equations (2) and (3), we have:(4)$$E({\vec{r}}^{\prime },t)=\frac{\exp(ikz)}{i\lambda z}\exp[\frac{i\pi }{\lambda z}{{\vec{r}}^{\prime }}^{2}]\int E(\vec{r},t-z/c)\exp[\frac{i\pi }{\lambda z}{\vec{r}}^{2}]\exp[-\frac{i\pi }{\lambda z}2\vec{r}\cdot {\vec{r}}^{\prime }]d\vec{r}.$$

Furthermore, the integral term on the righthand side can be regarded as the Fourier transform of $E(\vec{r},t-z/c)\exp[i\pi {\vec{r}}^{2}/\lambda z]$.

A computational ghost imaging system is demonstrated in Figure 3. We assume that the reflectivity of the object is $T({\vec{r}}^{\prime })$, the distance between the object and SLM is *z*_1_, and the distance between the detector and object is *z*_2_. According to Equation (2), the light field distribution on the object surface is:(5)$$E_{1}({\vec{r}}^{\prime },t)=\int E(\vec{r},t-z_{1}/c)h(\vec{r},{\vec{r}}^{\prime })d\vec{r}.$$

The light field distribution at the bucket detector is:(6)$$E_{1}({\vec{r}}^{\prime \prime },t)=\int E_{1}({\vec{r}}^{\prime },t-z_{2}/c)T({\vec{r}}^{\prime })h({\vec{r}}^{\prime },{\vec{r}}^{\prime \prime })d{\vec{r}}^{\prime },$$
thus, the measurement of the bucket detector is:(7)$$I_{1}(t)=\int E_{1}({\vec{r}}^{\prime \prime },t)E_{1}^{*}({\vec{r}}^{\prime \prime },t)d{\vec{r}}^{\prime \prime }.$$

While, the intensity of the light fields at distance $z_{1}+z_{2}$ is:(8)$$I_{2}({\vec{r}}^{\prime \prime },t)=E_{2}({\vec{r}}^{\prime \prime },t)E_{2}^{*}({\vec{r}}^{\prime \prime },t),$$
where
(9)$$E_{2}({\vec{r}}^{\prime \prime },t)=\int E(\vec{r},t-\left(z_{1}+z_{2}\right)/c)h(\vec{r},{\vec{r}}^{\prime \prime })d\vec{r}.$$

Then, the reconstructed image of the target object can be calculated by intensity correlation:(10)$$G\left({\vec{r}}^{\prime \prime }\right)=\langle I_{1}\left(t\right)I_{2}\left({\vec{r}}^{\prime \prime },t\right)\rangle -\langle I_{1}\left(t\right)\rangle \langle I_{2}\left({\vec{r}}^{\prime \prime },t\right)\rangle ,$$
where 〈〉 represents the ensemble average.

## 4. Numerical Simulation

We conduct our simulations of a computational ghost imaging in a remote sensing scenario. The object to be imaged, in this paper, is a double slit. The power of the light beam reflected from each light point pixel on the SLM is equal, the radius of the light beam is 0.3 mm, and the wavelength is 1064 nm. The polarization is linear. The modulated phase of each light point pixel is independent. Meanwhile, the phases obey uniform distribution in (−π,π]. Each light point pixel of the SLM can be regarded as a sub-light source, and the distance between two adjacent light point pixels is $d=\left(\sqrt{7}\times 0.04\;m\right)/18=0.0059\;m$. The area of the SLM is $0.04\times 0.04{\;m}^{2}$, and the area of the detector is $12\times 12{\;m}^{2}$. The size of the reconstructed image is 128×128 pixels. The propagation distance is 10 km.

### 4.1. Image Quality Evaluation Index

#### 4.1.1. Resolution

In the optical imaging system, the width of the point-spread function (PSF) determines the imaging resolution [32]. The width of the PSF in a computational ghost imaging system can be written as:(11)$$\Delta PSF=\frac{\lambda z}{\sqrt{2}\pi \rho_{s}},$$
where $\rho_{s}$ is the transverse size of the light source and *z* represents the distance between the SLM and detector.

The width of the PSF mentioned above is only suitable for characterizing the theoretical resolution of the optical imaging system. In remote sensing, the value of *z* is not easy to measure, so we consider using the full width at half maximum (FWHM) of the image pixel value to represent the resolution. When the light point pixel on the SLM is arranged according to the coprime array, part of the profile of the reconstructed image of the double slit is shown in Figure 4. We calculate the value of FWHM in the reconstructed image by the following steps: First, average the pixel values of all rows in the region where the double slits are located, and then calculate the distance between two pixels with a pixel value of 0.5, which is the FWHM value of a single slit. Finally, it is necessary to average the FWHM values of the left and right single slits. When the pixel value of the intersection of the two peaks is greater than 0.5, the target object cannot be distinguished. In this case, we image the left and right slits separately, then, calculate the FWHM values of the two single slits in turn, and finally, take the average to get the required FWHM value.

#### 4.1.2. Peak Signal-to-Noise Ratio

In order to quantify the performance of the recovered image, we denote the peak signal-to-noise ratio (PSNR) as follows:(12)$$PSNR=10\log_{10}\left(\frac{MAX_{I}^{2}}{MSE}\right),$$
where $MAX_{I}$ is the maximum pixel value of a picture, or 255 if each pixel is represented by an 8-bit binary, and $MSE\left(I,O\right)=\frac{1}{MN}\sum_{x=1}^{M}\sum_{y=1}^{N}{\left(I\left(x,y\right)-O\left(x,y\right)\right)}^{2}$ is the mean square error.

#### 4.1.3. Contrast-to-Noise Ratio

In this paper, the contrast-to-noise ratio (CNR) is defined as:(13)$$CNR=\frac{\frac{1}{N_{1}}\sum_{O\left(x\right)=1}I^{2}\left(x\right)-\frac{1}{N_{2}}\sum_{O\left(x\right)=0}I^{2}\left(x\right)}{\sqrt{\sigma_{1}^{2}+\sigma_{0}^{2}}},$$
where $\sum_{O\left(x\right)=1}I^{2}\left(x\right)$ is the sum of the pixel intensity corresponding to the light-transmitting area of the measured object in the reconstructed image, $\sum_{O\left(x\right)=0}I^{2}\left(x\right)$ is the sum of the pixel intensity corresponding to the opaque area of the measured object in the reconstructed image, *N*_1_ and *N*_2_ represent the total number of pixels in these two regions, $\sigma_{1}$ and $\sigma_{0}$ represent the pixel standard deviation of these two regions.

### 4.2. Comparison of Imaging Qualities of the Coprime Array, Uniform Array, and Random Sparse Array

Because the coprime array constructed in this paper contains 49 pixels (see Figure 2), in order to compare the imaging qualities of different arrays, the random sparse array and uniform array should also contain 49 pixels. In addition, although the increase of sampling number improves the quality of the reconstructed image, too many samples greatly increase the computation time. We found that a good imaging quality could be achieved when the sampling number was larger than 10% of the total number of pixels in the reconstructed image. In our experiment, we choose the sampling number as 2048, which was 12.5% of the total number. The resulting images are shown in Figure 5. The PSNR and CNR of the images are given in Table 1.

From Figure 5, we find that the image formed by a random sparse array has the largest background noise, while the image formed by a uniform array has periodical patterns. The period of the image formed by a uniform array is:(14)$$T=\frac{\lambda z}{d}$$
where *λ* is the wavelength of the light, *z* is the propagation distance, and *d* is the distance between two adjacent light point pixels. This periodicity interferes with our recognition of objects. For the image formed by the coprime array, on the one hand, it overcomes the disadvantage of large background noise in random sparse array ghost imaging, and makes the image of the object clearer than the background; on the other hand, it avoids the periodicity of uniform array imaging and facilitates the recognition of objects.

### 4.3. The Effect of the Distance between Adjacent Light Point Pixels on Imaging Noise of Coprime Array

Let the distance between the adjacent light point pixels in **H_1_** be *d*_1_ and the distance between the adjacent light point pixels in **H_2_** be *d*_2_. According to the previous discussion, we have $d_{1}=d_{2}=d=0.0059\;m$. Because a single subarray can be regarded as a uniform array, the imaging of a single subarray is periodic, and the period is:(15)$$T=\frac{\lambda z}{d}=\frac{1064\;\mathrm{nm}\times 10\;\mathrm{km}}{0.0059\;m}=1.8097\;m$$

According to Figure 5, we find that the distribution of background noise has a certain periodicity, and we think that the periodic distribution of noise is caused by the periodicity of a single subarray imaging. Therefore, changing the distance between adjacent light point pixels affects the imaging period of a single subarray, and then affects the distribution of background noise.

Keeping other conditions unchanged, the effect of the distance between adjacent light point pixels on the imaging background noise is verified by changing the values of *d*_1_ and *d*_2_, and the resulting image is shown in Figure 6.

As shown in Figure 6, the noise of the images corresponding to $d_{1}=d/2,d_{2}=d$ and $d_{1}=d,d_{2}=d/2$ happens to be in a complementary position, because when $d_{1}=d/2$ ($d_{2}=d/2$), the period of noise caused by **H_1_** (**H_2_**) becomes twice as long as the original, and part of the noise does not appear in the image. When $d_{1}=d/2$ and $d_{2}=d/2$, the noise in the image is significantly reduced because the period of the noise caused by the two subarrays is doubled.

Then, the effect of the distance between adjacent light point pixels on the imaging noise of the coprime array is analyzed by the two indexes of PSNR and CNR. According to the data in Table 2, when the distance between adjacent light point pixels becomes smaller, the values of PSNR and CNR increase significantly, which is mainly caused by the reduction of background noise.

### 4.4. The Effect of the Distance between Adjacent Light Point Pixels on Imaging Periodicity of Uniform Array

According to Equation (14), decreasing the value of *d* increases the imaging period of a uniform array. When *d* decreases to a certain value (*T* increases to a certain value), there is only one period in the reconstructed image. We consider a uniform array containing n×n light point pixels, n=7,8,9,⋯. While keeping the size of the SLM unchanged, increasing the value of *n* decreases the value of *d*. We find from the simulation results (see Figure 7) that when the value of *n* increases from 12 to 13, the imaging quality is significantly improved.

When n=11,12,13,14, the values of PSNR and CNR of the uniform array imaging are shown in Table 3. A comparison with the values of coprime array in Table 1 shows that only when n≥13, is the value of each index of the uniform array better than that of the coprime array containing 49 light point pixels. This also verifies that the imaging quality of the coprime array is better than that of the uniform array when the number of light point pixels is the same.

### 4.5. The Effect of the Transverse Size of Light Source on Imaging Resolution

According to Equation (11), when the transverse size of the light source becomes smaller, the width of the point spread function becomes larger, which leads to poor resolution of the reconstructed image. Figure 8 shows the image and its profile formed by a coprime array containing 49 light point pixels when the transverse dimensions of the light source are $\rho_{s}=0.04\;m$ and $\rho_{s}=0.02\;m$, whereas Figure 9 and Figure 10 show the image and its profile formed by a uniform array and a random sparse array under the same conditions, respectively.

It can be found from Figure 8, Figure 9 and Figure 10 that when the transverse size of the light source changes to half of the original, the edge of the image becomes blurred, and the imaging resolution becomes worse. The values of FWHM, PSNR, and CNR are shown in Table 4. The value of FWHM increases with a decrease of the transverse size of the light source. When the transverse size of the light source is 0.04 m, the FWHM values of the three arrays are almost the same, whereas when the transverse size of the light source is 0.02 m, the FWHM value of the coprime array is significantly higher than that of the random sparse array and the uniform array.

In addition, as shown in Figure 8, Figure 9 and Figure 10 and Table 4, when the transverse size of the light source becomes half of the original, the background noise of the three kinds of array imaging is significantly reduced. The above phenomenon is mainly because the decrease of the transverse size of the light source is equivalent to the decrease of the distance between the adjacent light point pixels without changing the number of light point pixels. According to the discussion in Section 4.3, it is known that when the distance between the adjacent light point pixels becomes smaller, the period of noise becomes larger, and part of the noise does not appear in the image. Additionally, when the transverse size of the light source is the same, the image formed by the coprime array overcomes the disadvantage of large background noise in random sparse array imaging, and also avoids the periodicity of uniform array imaging.

## 5. Conclusions

In this paper, we introduce a coprime array to optimize the arrangement of light point pixels on an SLM used in computational ghost imaging. At present, the commonly used arrangement of light point pixels is uniform arrangement and random sparse arrangement. When light point pixels are uniformly arranged, more light point pixels are needed to achieve better imaging quality, and too few light point pixels cause periodic patterns resulting in target aliasing; when light point pixels are randomly and sparsely arranged, the images suffer from large background noise. Through the numerical simulation, we found that the reconstructed image obtained by the coprime array had less background noise and avoided the periodicity of imaging. When using a coprime array, the values of PSNR and CNR were also the best. We also discussed the influence of the distance between the adjacent light point pixels and the transverse size of the light source on the imaging quality.

## Figures and Tables

**Figure 1 sensors-20-04508-f001:**
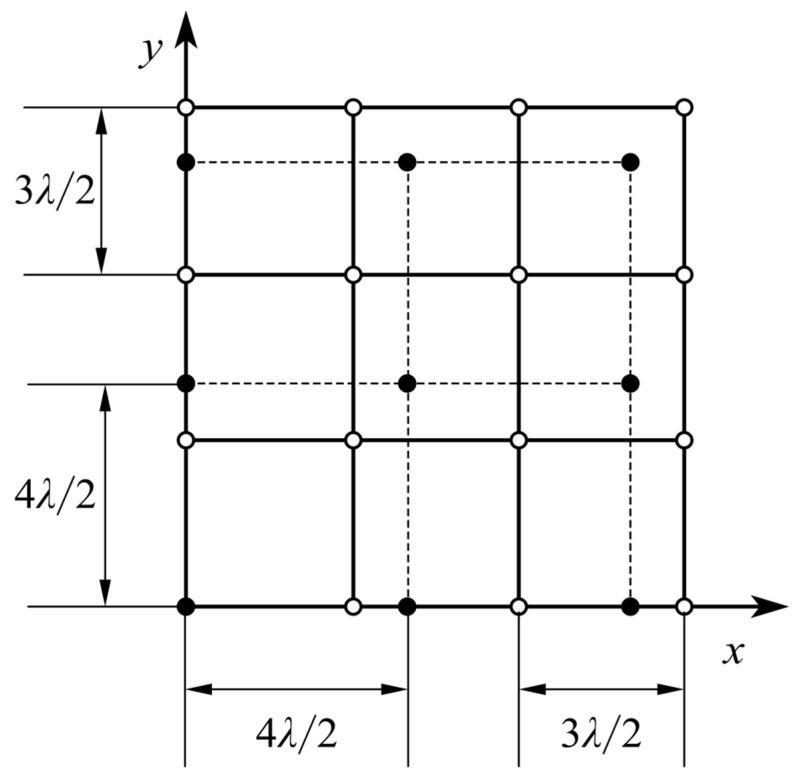
Example of coprime planar array with M=3 and N=4.

**Figure 2 sensors-20-04508-f002:**
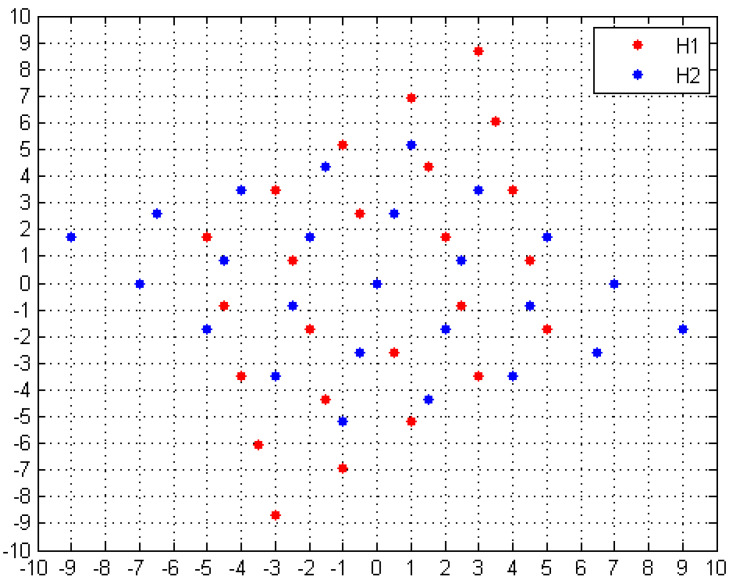
Coprime array constructed on an Eisenstein domain.

**Figure 3 sensors-20-04508-f003:**
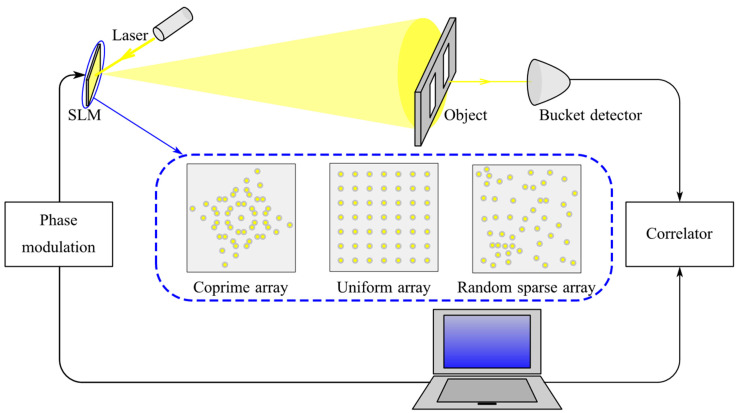
Schematic diagram of a computational ghost imaging system.

**Figure 4 sensors-20-04508-f004:**
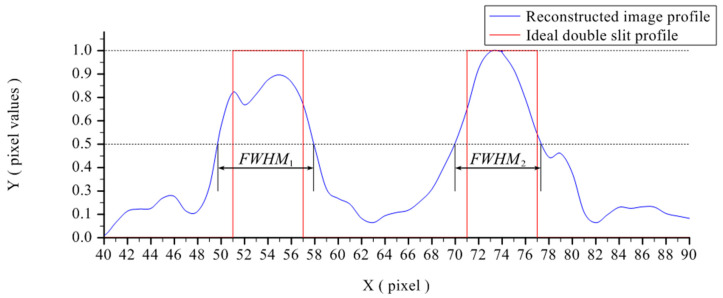
Partial profile of a double-slit reconstructed image.

**Figure 5 sensors-20-04508-f005:**
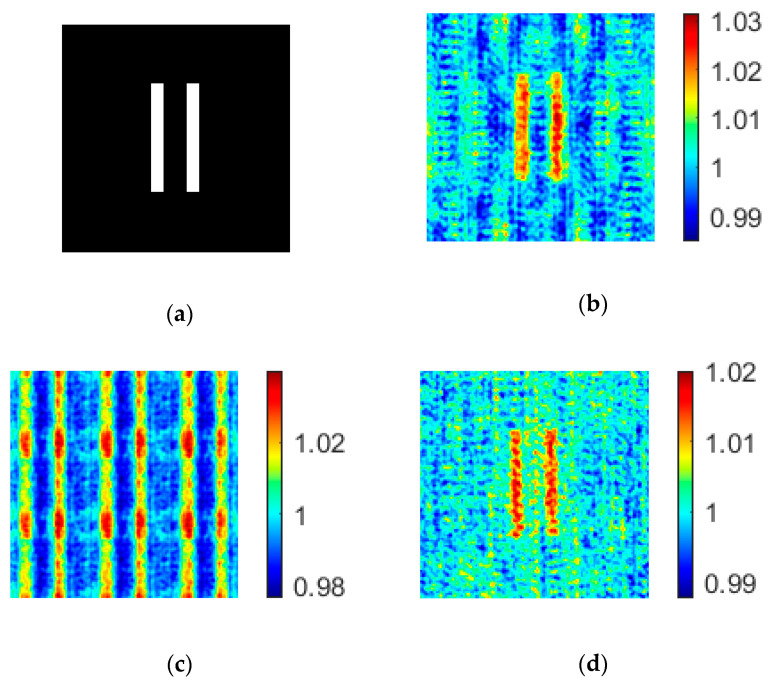
Images formed by coprime array, uniform array, and random sparse array with the same number of light point pixels. (**a**) Object; (**b**) Imaging results based on coprime array; (**c**) Imaging results based on uniform array; (**d**) Imaging results based on random sparse array.

**Figure 6 sensors-20-04508-f006:**
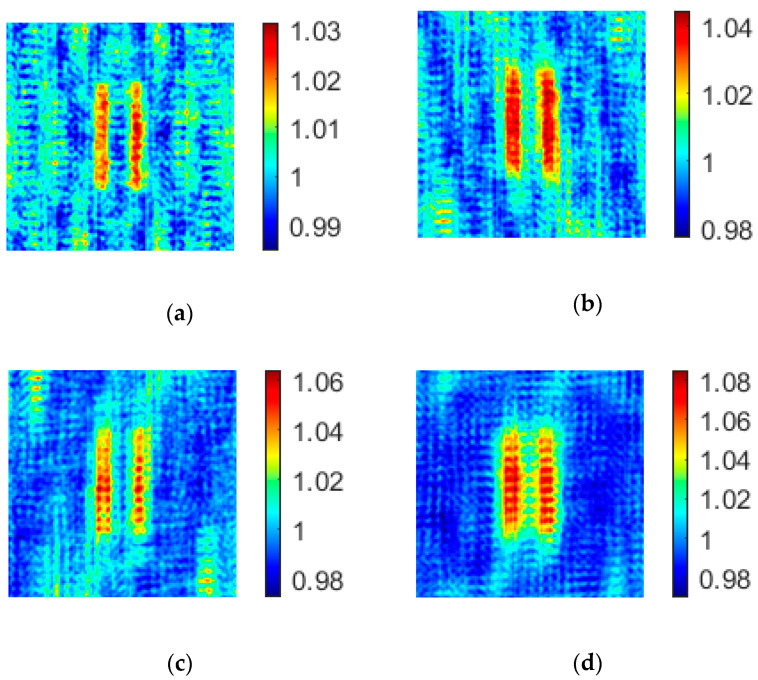
The effect of the distance between adjacent light point pixels on imaging noise of a coprime array. (**a**) $d_{1}=d_{2}=d=0.0059\;m$; (**b**) $d_{1}=d/2,d_{2}=d$; (**c**) $d_{1}=d,d_{2}=d/2$; (**d**) $d_{1}=d/2,d_{2}=d/2$.

**Figure 7 sensors-20-04508-f007:**
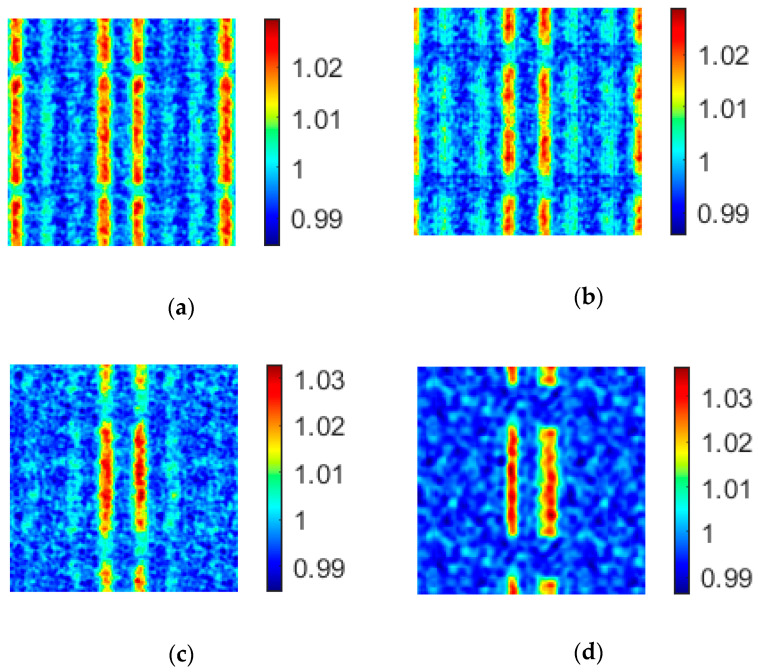
The effect of the distance between adjacent light point pixels on imaging periodicity of n×n uniform array. (**a**) n=11; (**b**) n=12; (**c**) n=13; (**d**) n=14.

**Figure 8 sensors-20-04508-f008:**
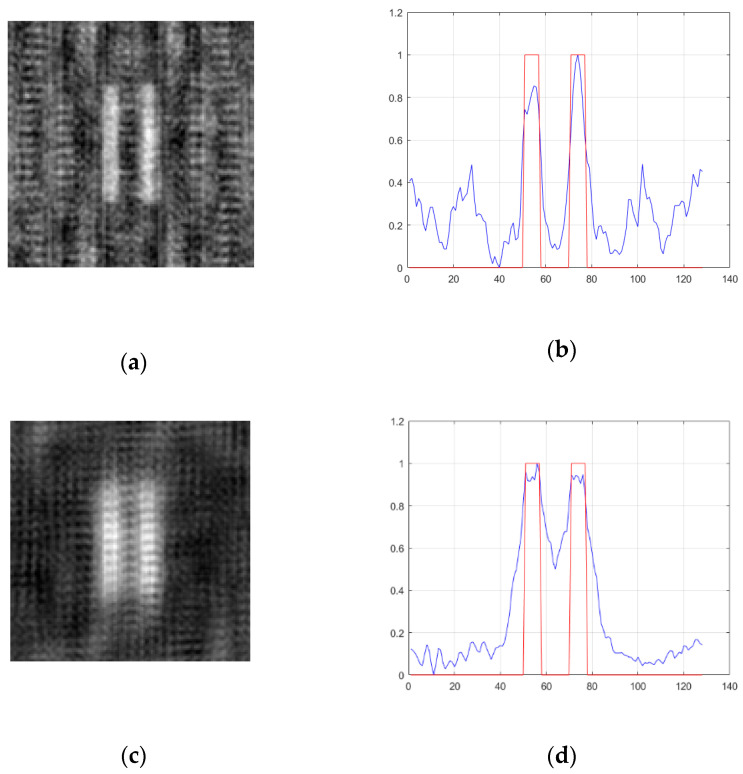
The effect of the transverse size of light source on imaging resolution of a coprime array. (**a**,**b**) $\rho_{s}=0.04\;m$; (**c**,**d**) $\rho_{s}=0.02\;m$. Left, reconstructed image; right, section view of reconstructed image. The red line represents the double slit and the blue line represents the reconstructed image of the double slit.

**Figure 9 sensors-20-04508-f009:**
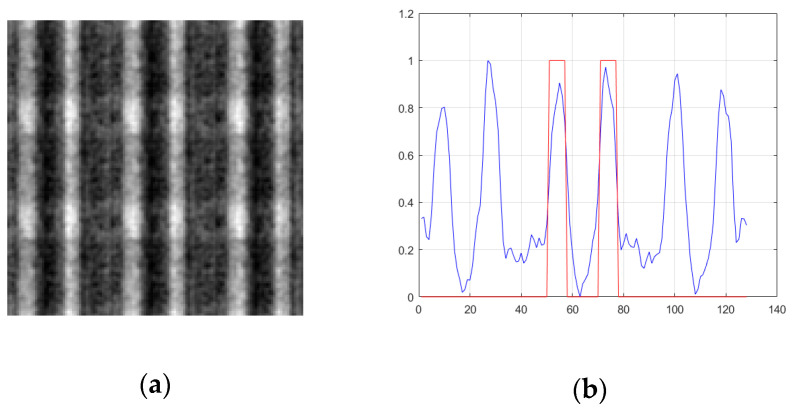
The effect of the transverse size of light source on imaging resolution of a uniform array. (**a**,**b**) $\rho_{s}=0.04\;m$; (**c**,**d**) $\rho_{s}=0.02\;m$. Left, reconstructed image; right, section view of reconstructed image. The red line represents the double slit and the blue line represents the reconstructed image of the double slit.

**Figure 10 sensors-20-04508-f010:**
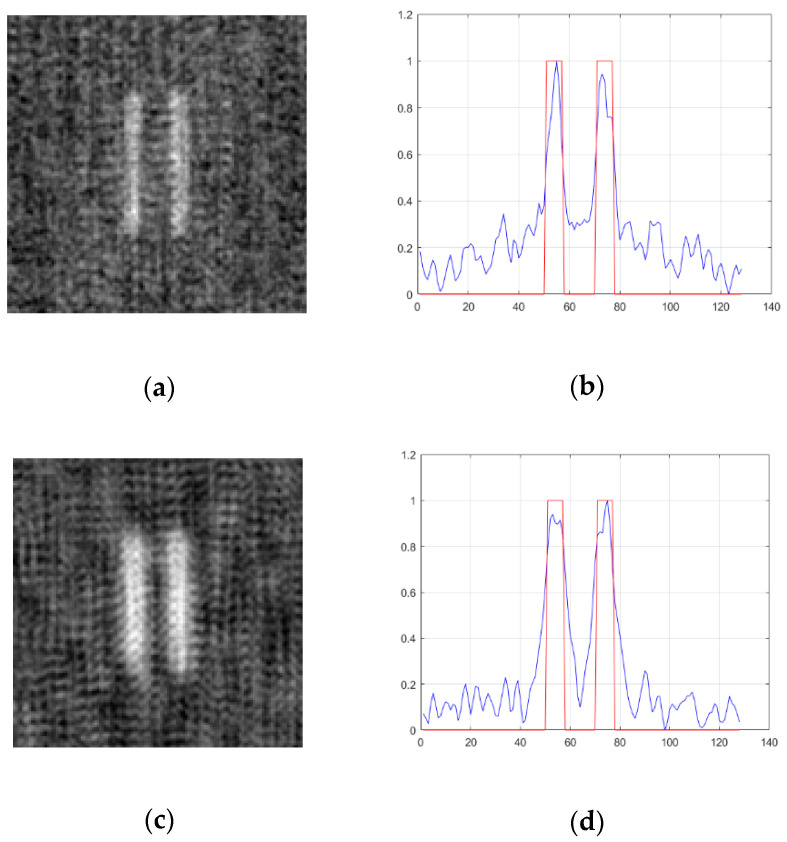
The effect of the transverse size of light source on imaging resolution of a random sparse array. (**a**,**b**) $\rho_{s}=0.04\;m$; (**c**,**d**) $\rho_{s}=0.02\;m$. Left, reconstructed image; right, section view of reconstructed image. The red line represents the double slit and the blue line represents the reconstructed image of the double slit.

**Table 1 sensors-20-04508-t001:** The values of peak signal-to-noise ratio (PSNR) and contrast-to-noise ratio (CNR) for different light source array structures.

	Coprime Array	Uniform Array	Random Sparse Array
PSNR	9.2019	7.6906	8.2299
CNR	2.4898	1.3345	2.3451

**Table 2 sensors-20-04508-t002:** The effect of the distance between adjacent light point pixels on imaging noise of a coprime array.

	$d_{1}=d_{2}=d$	$d_{1}=d/2,d_{2}=d$	$d_{1}=d,d_{2}=d/2$	$d_{1}=d/2,d_{2}=d/2$
PSNR	9.2019	9.4754	10.4300	11.2744
CNR	2.4898	3.2052	2.7857	3.1331

**Table 3 sensors-20-04508-t003:** The effect of the distance between adjacent light point pixels on imaging periodicity of n×n a uniform array.

	*n* = 11	*n* = 12	*n* = 13	*n* = 14
PSNR	7.7302	7.7887	10.9989	11.5531
CNR	1.6165	1.9883	2.8707	2.9713

**Table 4 sensors-20-04508-t004:** The effect of transverse size of light source on imaging resolution.

		PSNR	CNR	FWHM
Coprime array	$\rho_{s}=0.04\;m$	9.2019	2.4898	7.52
	$\rho_{s}=0.02\;m$	11.2744	3.1331	16.24
Uniform array	$\rho_{s}=0.04\;m$	7.6906	1.3345	7.21
	$\rho_{s}=0.02\;m$	10.9489	2.9432	13.19
Random sparse array	$\rho_{s}=0.04\;m$	8.2299	2.3451	7.15
	$\rho_{s}=0.02\;m$	10.4247	3.1978	10.74

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
