# Peer review of "Computational Ghost Imaging Based on Light Source Formed by Coprime Array"

_sensors, 2020, doi:10.3390/s20164508_

Round 1

Reviewer 1 Report

The authors proposed to apply coprime arrays for computational ghost imaging patterns. The idea was to address the problem of frequency aliasing in periodic sampling and low SNR in random sampling. However, the reasoning of the proposition is not well-explained, the quality of the result is poor and not convincing, given that the results were yielded from numerical simulation. Having said that, I cannot recommend the manuscript for publication before the following comments are properly addressed.

  1. In Introduction, the authors claimed that ‘in order to obtain better image quality, a large number of light point pixels are needed’, which is not necessarily true. If one samples the scene with orthogonal patterns, then only N measurements are needed for an N pixel image to yield the perfect result in theory or numerical simulation. I suggest the authors refer to recent reviews, such as Single-pixel imaging and its application in three-dimensional reconstruction: a brief review. Sensors, 19, 732, 2019, so that both the authors and the potential readers could have a better understanding of computational ghost imaging and single-pixel imaging.

  1. The authors claimed that they ‘the arrangement of light point pixels on SLM is optimized based on coprime array’. However, they didn’t explain why such array will optimize the sampling pattern in computational ghost imaging, just mentioned that such array is used in microwave radar.
  1. The result illustrated in Fig. 5c only happened because the sampling pattern is periodic and the sampling number is small, and therefore formed an ill-posed math problem which cannot determine the location of the object inside the field of view. In the case when the object is on the corner of the FOV, then the coprime array will yield no image at all because there is not sampling point in the corner of the FOV for coprime array. This example is to demonstrate that the comparison is not a fair one.

  1. The results are not convincing in three ways. First, it had only one object. Second, the object is very simple. Third, it is only numerical. I think for numerical simulation, the authors should not be limited to such poor results.

Reviewer 2 Report

This is a generally well written paper concerning computational ghost imaging, and comparing the results obtained by simulation using spatial light modulator patterns based on uniform, sparse random, and coprime two dimensional arrays. Good results are obtained but the analysis measures used require consideration and justification.

The english grammar and prose is of a good standard but needs some correction to improve repeated grammatical errors.

Specific points noted which require correction/modifiaction are:

Line 27: Incorrect usage of verb tenses.

Line 56, S2: Improve section title

Line 61: This is the first mention of wavelength. Clearly explain what this relates to.

Page 3: Improve formatting by eliminating in-line equations involving matrices. Use separate lines for matrix equations.

Line 71 to 98: The explanation of the generation of the coprime array based on an Eisenstein domain requires more detail and all terms to be defined, if it is to be included in such depth. The method is already referenced i.e. [26] so it could, alternatively, be summarized more simply.

Line 111: There's no verb in this sentence.

Line 140: "z" - remove superscript

Line 153/S4.1.2, Line 158/S4.1.3, and line 208: It's not clear how sensitive either PSNR or CNR are to the presence or absence periodic patterns produced, particular, by the uniform array. Is there a more suitable figure of merit for periodic features?

Figures 5, 6 and 7: Include a labelled colour bar so the reader can see the contrast depth.

Line 191: Use a space between values and units and don't italicize the units - and all other value/units items.

Figure 6 caption is orphaned from the figure - keep together.

Entire document: Indefinite and definite articles missing in numerous places.

Round 2

Reviewer 1 Report

I think the authors addressed the comments with satisfying modifications, therefore I recommend the work for publication